# High Diversity of *Fusarium* Species in Onychomycosis: Clinical Presentations, Molecular Identification, and Antifungal Susceptibility

**DOI:** 10.3390/jof9050534

**Published:** 2023-04-30

**Authors:** Lai-Ying Lu, Jie-Hao Ou, Rosaline Chung-Yee Hui, Ya-Hui Chuang, Yun-Chen Fan, Pei-Lun Sun

**Affiliations:** 1Department of Dermatology, Chang Gung Memorial Hospital, Linkou Branch, Taoyuan 333423, Taiwan; 2College of Medicine, Chang Gung University, Taoyuan 333323, Taiwan; 3Department of Dermatology and Aesthetic Medicine Center, Jen-Ai Hospital, Taichung 412224, Taiwan; 4Department of Plant Pathology, National Chung Hsing University, Taichung 402202, Taiwan; 5Research Laboratory of Medical Mycology, Chang Gung Memorial Hospital, Linkou Branch, Taoyuan 33323, Taiwan

**Keywords:** non-dermatophyte mould, *Fusarium* onychomycosis, histopathology in onychomycosis

## Abstract

*Fusarium* are uncommon but important pathogenic organisms; they cause non-dermatophyte mould (NDM) onychomycosis. Patients typically respond poorly to treatment owing to Fusarium’s native resistance to multiple antifungal drugs. However, epidemiological data for Fusarium onychomycosis are lacking in Taiwan. We retrospectively reviewed the data of 84 patients with positive *Fusarium* nail sample cultures at Chang Gung Memorial Hospital, Linkou Branch between 2014 and 2020. We aimed to investigate the clinical presentations, microscopic and pathological characteristics, antifungal susceptibility, and species diversity of *Fusarium* in patients with Fusarium onychomycosis. We enrolled 29 patients using the six-parameter criteria for NDM onychomycosis to determine the clinical significance of *Fusarium* in these patients. All isolates were subjected to species identification by sequences and molecular phylogeny. A total of 47 Fusarium strains belonging to 13 species in four different *Fusarium* species complexes (with *Fusarium keratoplasticum* predominating) were isolated from 29 patients. Six types of histopathology findings were specific to Fusarium onychomycosis, which may be useful for differentiating dermatophytes from NDMs. The results of drug susceptibility testing showed high variation among species complexes, and efinaconazole, lanoconazole, and luliconazole showed excellent in vitro activity for the most part. This study’s primary limitation was its single-centre retrospective design. Our study showed a high diversity of *Fusarium* species in diseased nails. Fusarium onychomycosis has clinical and pathological features distinct from those of dermatophyte onychomycosis. Thus, careful diagnosis and proper pathogen identification are essential in the management of NDM onychomycosis caused by *Fusarium* sp.

## 1. Introduction

*Fusarium* is a widely distributed hyaline mould genus with at least 300 phylogenetically different species in 23 species complexes [1]. These species have notable human pathogenicity; however, despite their diversity, only a few, such as *F. solani* species complex (FSSC), *F. oxysporum* species complex (FOSC), and *F. fujikuroi* species complex (FFSC), cause disease in humans [2]. In immunocompetent patients, locally invasive onychomycosis and keratitis are the most frequent manifestations; however, in immunocompromised patients, severe disseminated disease may cause mortality [3]. Owing to late diagnosis, intrinsic resistance to azole antifungals, and the emergence of multidrug resistant strains due to agricultural antifungal overuse, treatment of fusariosis is a major challenge. 

Fusarium onychomycosis accounts for 0.97–6% of cases of onychomycosis [4], and *Fusarium* is a causative agent in 9–44% of cases of non-dermatophyte mould (NDM) onychomycosis [5]. *Fusarium* is a common environmental and agricultural fungus which could be a contaminant of clinical nail samples. Therefore, repeated culture may be required to determine the true pathogenicity. Trauma, soil contact, and walking barefoot are the primary causes of Fusarium onychomycosis, which preferentially affects the big toe with clinical phenotypes of superficial, subungual, or acute paronychia. Clinically important *Fusarium* species are typically resistant to all antifungals including azoles, echinocandins, and polyenes [6]. This, along with the diversity of pathogenic *Fusarium* species, highlights the importance of fungal culture and molecular identification [7]. 

Climate, environmental, and socioeconomic factors may also have affected the epidemiological profiles of causal onychomycosis agents throughout time [8]. However, there remains a lack of studies regarding the clinicopathological and epidemiological characteristics of Fusarium onychomycosis in Taiwan. Thus, we aimed to investigate the clinical presentations, microscopic and pathological characteristics, and *Fusarium* species diversity in patients with Fusarium onychomycosis.

## 2. Materials and Methods

We retrospectively reviewed data obtained from 84 patients with positive nail cultures of *Fusarium* at the Chang Gung Memorial Hospital, Linkou branch, between 2014 and 2020. Demographic data, history of soil contact, associated predisposing factors (including diabetes mellitus, malignancy, and an immunocompromised status), treatment (topical or systemic antifungals, surgery, and laser therapy), and prognosis were collected from medical records and by telephone interview. Photographs of the affected nails were taken with patient consent, and this study was reviewed and approved by the Institutional review board (IRB) of the Chang Gung Medical Foundation (approval number 202101575B0). Patient consent was waived by the IRB.

Diagnosis of Fusarium onychomycosis was made based on the six-parameter criteria for NDM onychomycosis proposed by Gupta et al., as follows: (1) identification of NDMs by microscopy using potassium hydroxide (KOH) preparation, (2) culture isolation of NDMs, (3) repeated isolation of the same NDM in culture, (4) failure to isolate a dermatophyte in culture, (5) culture of the same NDM from 5 out of 20 inoculations of nail fragments, and (6) NDM identification using molecular techniques or histological findings [9]. In this study, we made a diagnosis of Fusarium onychomycosis when parameter (4) and at least two to three other parameters were fulfilled.

### 2.1. Sample Collection, Culture, and Microscopic/Histopathological Examination

Diseased portions of subungual nail debris or plates were collected using a nail clipper or scalpel. Debris was pre-treated with 20% KOH and examined by microscopy to detect fungal elements. Nail plates were sent for histopathological examination and stained with haematoxylin and eosin and Periodic acid–Schiff stains. For fungal culturing, nail debris was inoculated on both inhibitory mould (CMP^®^, Creative Life Sciences, Taipei, Taiwan) and Mycosel agar (BD Difco™, BD, Franklin Lakes, NJ, USA) plates and incubated at 25 °C. The fungus grown was purified by subculture on Sabouraud’s dextrose agar plates (BD Difco™) for morphological identification and molecular study.

### 2.2. Molecular Identification 

All *Fusarium* isolates were subjected to sequence-based molecular identification. The fungal genomic DNA was obtained using the Smart LabAssist (TANBead, TANBead, Taoyuon City, Taiwan) automatic DNA extraction machine. The internal transcribed spacers (ITS) of ribosomal DNA were amplified with primers—ITS1 (TCCGTAGGTGAACCTGCGG) and ITS4 (TCCTCCGCTTATTGATATGC); the partial transcription elongation factor-1α (*TEF-1α*) gene was amplified with primers EF1 (ATGGGTAAGGARGACAAGAC) and EF2 (GGARGTACCAGTSATCATG). Polymerase chain amplification (PCR) products were confirmed by electrophoresis, purified, and sequenced using an ABI Prism 3730 xl DNA analyser (Applied Biosystems, Foster City, California, USA). Sequences generated in this study were deposited at the DNA Data Bank of Japan (DDBJ) [10]. Preliminary identification performed by comparing the sequences of each *Fusarium* isolate with sequences deposited in the Fusarium MLST (Multilocus Sequence Typing) database at the Mycobank website (https://fusarium.mycobank.org/ (accessed on 15 February 2022)) and the *Fusarium* Database (http://isolate.fusariumdb.org (accessed on 15 February 2022)). Identification was confirmed by phylogenetic analysis. 

Based on preliminary identification results from the *Fusarium* MLST database, sequences of *Fusarium* species similar to the strains used in this study were downloaded from GenBank (https://www.ncbi.nlm.nih.gov/genbank/ (accessed on 15 June 2022)). *Atractium crassum* was selected as the outgroup for subsequent analysis, and sequences were first aligned by multiple alignment using fast Fourier transform (online version; https://mafft.cbrc.jp/alignment/server/ (accessed on 15 June 2022)). During manual inspection, any poorly aligned regions were removed using Gblocks [11]. Finally, the *TEF-1α* and ITS regions were concatenated for subsequent analyses. A maximum-likelihood tree was generated using the workflow in IQ-TREE 2.1.3 [12], and DNA models were automatically selected by the built-in ModelFinder algorithm [13]. The support value of the nodes was calculated from 1000 repeated slow standard nonparametric bootstrap. A Bayesian inference tree was obtained by analysing the same dataset with MrBayes v3.2.6 [14]. The analysis started with two MCMC chains of 1,000,000 generations, and one tree was kept every 1000 generations. The last three quarters of the 1000 trees obtained were used to compute the final consensus tree, and trees were visualized using MEGA 7 [15]. All analyses were performed using a Linux Mint 20.3 (64-bit) operating system, and to ensure reproducibility, random seeds were explicitly set to 56 wherever necessary.

### 2.3. Antifungal Susceptibility 

Forty-seven specimens were included from 29 patients with positive fungal culture. Drug susceptibility tests were performed in accordance with the third edition of M38: Reference Method for Broth Dilution Antifungal Susceptibility Testing of Filamentous Fungi, published by the Clinical and Laboratory Standards Institute [16]. *Candida parapsilosis* ATCC 22019, *C. krusei* ATCC 6258, and *Trichophyton mentagrophytes* ATCC MYA-4439 were used as controls. Antimicrobial agents, and the range of concentration tested included amphotericin B (AMB; 16–0.031 μg/mL), terbinafine (TRB; 32–0.063 μg/mL), fluconazole (FLC; 64–0.125 μg/mL), itraconazole (ITC; 32–0.063 μg/mL), efinaconazole (EFC; 4–0.008 μg/mL), lanoconazole (LNC; 4–0.008 μg/mL), luliconazole (LLC; 4–0.008 μg/mL), voriconazole (VRC; 16–0.031 μg/mL), and natamycin (NAT; 32–0.063 μg/mL). Minimal inhibitory concentration (MIC) for all antifungals to the fungal isolate were determined by 100% mycelium growth inhibition following 48 h of incubation at 35 °C.

## 3. Results 

*Fusarium* was isolated from the nail samples of 84 patients, 55 of whom did not fulfil the criteria for diagnosis of NDM onychomycosis and were excluded. Finally, 29 patients were enrolled for further analysis. 

### 3.1. Demographic Data and Clinical Manifestations 

After rigorous clinical, histological, and mycological confirmation, 29 patients were diagnosed with Fusarium onychomycosis (Table 1). There were 13 men (44.8%) and 16 women (55.2%), with the mean age of 55 (3–87) years. Average disease duration was 20 (1–108) months. Six (20.7%) patients had a personal history of gardening or soil contact, and seven (24.1%) had diabetes (*n* = 2), immunocompromise (*n* = 2), and underlying cancer (*n* = 3) as predisposing factors. 

The most commonly involved nails were toenails (*n* = 16), especially the first toe (*n* = 15); however, there was one case in bilateral thumbs and six in fingernails. Six patients had both fingernail and toenail onychomycosis; however, none of the six had predisposing factors of diabetes, immunocompromise, or cancer. Clinical manifestations of Fusarium onychomycosis differed from those of classic dermatophyte onychomycosis; they included yellow to greenish discoloration, onycholysis, paronychia (mild or severe), and proximal subungual onychomycosis (PSO) (Figure 1). 

### 3.2. Direct Microscopic and Histological Findings 

Direct microscopic examination (DME) was performed on three patients, with frequent branching irregularly septated hyphae (Figure 2e), chlamydospore-like swelling (Figure 2f), terminal swelling and beading, and adventitious sporulation observed (Figure 2g). Among 29 patients, 21 had histopathological evidence of nail plate invasion. The histological characteristics of Fusarium onychomycosis could be categorized into six patterns, which provide clues for differentiating NDM onychomycosis from dermatophyte onychomycosis (Figure 3, Table 2): (1) presence of frequently branching irregularly septated hyphae, (2) arbitrarily widening hyphae, (3) dermatophytoma-like fungal mass, (4) thin hyphae embedded in nail specimen, (5) moniliform hyphae, and (6) hyphae with terminal swelling. The most encountered pathological finding was frequently branching irregularly shaped hyphae (Figure 3a), followed by moniliform hyphae (Figure 3e), and hyphae with terminal swelling (Figure 3f). 

### 3.3. Molecular Identification 

A total of 47 *Fusarium* strains were isolated, and the *TEF-1α* and ITS regions of all strains were successfully amplified and sequenced. The lengths of sequences obtained were 506–530 bp (DDBJ accession no. LC697741-LC697787) and 665–704 bp (DDBJ accession no. LC687503-LC687549). Based on phylogenetic analyses (Figure 4), the causative pathogens of confirmed cases of Fusarium onychomycosis were the FSSC (*n* = 23, including *F. keratoplasticum*, *F. falciforme*, *F. solani*, *Fusarium lichenicola*, *F. suttonianum*, *Nectria bolbophylli*, *Fusarium* sp.), *Fusarium incarnatum-equiseti* species complex (FIESC, *n* = 2, *F. arcuatisporum*, *F. pernambucanum*), FFSC (*n* = 3, *F. annulatum*, *F. denticulatum*), and FOSC (*n* = 2, *F. curvatum* and *Fusarium* sp.). One patient was infected by two *Fusarium* species at the same time (Figure 2).

### 3.4. Antifungal Susceptibility Testing

Antifungal susceptibility results are presented in Table 3. Obvious susceptibility differences between and within species complex were noted. In general, all isolates had very high MICs to FLC and ITC and low MICs to EFC, LNC, and LLC. FSSC had higher TRB MICs than non-FSSC, although FIESC had only one strain in the group, and the rough data are not so precise. The range of MICs are similar in VRC and NAT, and lower in AMB. Three *F. keratoplasticum* isolates were resistant to all antifungals, and one of them had a lower MIC to LNC. 

### 3.5. Treatment Response and Prognosis

Among 13 patients who received topical antifungal agents alone (Table 4), more than half (53.8%) had poor response, while two had good response. Nine patients received combination therapy (TRB and topical antifungal agents); however, more than half (55%) of them still had a poor response. Two patients received combination therapy (ITC and topical antifungal agents); one had a good response while the other had a poor response. Only three patients received TRB/ITC treatment with continuous topical antifungal agents, contributing to the good response observed in them. 

### 3.6. Presentation of Two Special Cases 

Case 1. A 60-year-old woman presented with yellow to greenish discoloration on her right 2nd finger, left thumb, and 2nd and 4th fingers for 2–3 months (Figure 2a,b). She had systemic lupus erythematosus (SLE) and was under treatment for oral azathioprine. She denied gardening history or contact with soil. Direct microscopic examination of the nail specimen demonstrated irregularly segmented hyphae with frequent branching, adventitious sporulation, chlamydospore-like swelling, and thin hyphae, which was different from that of dermatophytes (Figure 2e–g). Histopathology of the nail revealed septated hyphae and chlamydospores (Figure 2h). Fungal cultures from diseased nails all grew *Fusarium*, and molecular identification showed that her right thumb was infected by *F. keratoplascum* (CGMHD 0974), and her left thumb and 2nd and 4th fingers were infected by *F. solani* (CGMHD 0975, CGMHD 0976, CGMHD 0977). Repeated culture 3 months later also revealed the same results. The patient responded poorly to oral itraconazole (200 mg/day) for 3 months and was later treated with oral terbinafine (250 mg/day) combined with nail debridement and topical sulconazole solutions. A growth of new nails was noted three months later (Figure 2c,d). 

Case 2. A 55-year-old female patient had yellow to greyish discoloration on the bilateral big toes for several years (Figure 5a). She was a case of HBV chronic hepatitis and goiter of the thyroid. She denied gardening habit or contact to soil or other underlying disease, such as diabetes, malignancy, or under immunosuppressive treatment. Histopathology of the diseased nail demonstrated septated hyphae with a beaded appearance which invaded the nail plate (Figure 5b). Direct microscopic examination revealed septated hyphae and adventitious sporulation (Figure 5c). Six *Fusarium* isolates were cultured from the diseased nails during the two years of follow-ups. Molecular identification proved that all of them were *F. keratoplasticum*, but of three different genotypes based on the *TEF-1α* sequences (Figure 5d) The patient initially received oral griseofulvin 500 mg/day and topical antifungals for 21 days, but in vain. The patient received intermittent nail debridement and treatment with topical sulconazole solution in the following 3–4 years. New healthy nails finally grew with negative culture results. No recurrence was noted.

## 4. Discussion

Diagnosis of Fusarium onychomycosis is challenging because NDMs are common contaminants of nails. Published diagnostic criteria vary, and there is no consensus [9]. Approximately 42.8% false negative cases of NDM onychomycosis may be misdiagnosed when only negative dermatophyte microscopic examination and repeated culture are performed [17]. Gupta et al. proposed using three of their six clinical guideline criteria (KOH identification, isolation in culture, repeated isolation, inoculum counting [18], dermatophyte exclusion, and histological proof) to rule out dermatophyte contamination, and this remains the most widely used diagnostic method [8]. Although this method cannot perfectly prevent misdiagnosis of false negative and contaminants, it is straightforward and useful in aiding clinicians in clinical practice. For example, regarding inoculum counting, Gupta et al. had pointed out the low predictive value of inoculum counting as 23.2% of the time [19]. Similar histology finding may also be found in dermatophyte histology, but special appearance of dermatophytoma, irregulated septated hyphae, and terminal swelling are seldom seen in dermatophyte. The systemic review and comparison of histology difference in NDM and dermatophyte is important but still lacking, except in our clinical observations. When only using DME positive and negative dermatophyte culture for NDM diagnosis, there are only 53.6% sensitivity and 70.3% specificity [20]. Classical criteria include positive DME and repeated culture with 92.7% accuracy without the possibility of contaminants, but this method is difficult to be used in clinical practice [20]. Therefore, we follow the criteria of Gupta et al. and furthermore, 21 out of 29 patients in our research had histopathological evidence of fungal invasion with signs of NDM histology features, which can decrease rates of contaminants. New diagnostic methods, including molecular methods and techniques involving PCR, are seeing increasing application and importance. The commercialization of PCR kits may improve fungal diagnosis in the future [8]. 

In the clinical presentation of Fusarium onychomycosis, only 27.5% of patients had predisposing factors such as diabetes, immunosuppression, and cancers. The majority of *Fusarium* subtypes vary across the research, with proximal subungual onychomycosis (PSO), total dystrophic onychomycosis (TDO), and paronychia previously regarded as the most common clinical phenotypes [7,21]. However, distal lateral subungual onychomycosis (DLSO) and onycholysis are reportedly the more predominant subtypes [5,22,23,24,25]. FOSC is predominant in DLSO cases, and FSSC is more commonly involved in PSO, TDO, and SWO phenotypes according to Uemura et al. [25], but research from the north of Iran demonstrated diversity species among different subtypes of onychomycosis, with *F. proliferatum*, *F. keratoplasticum*, and *F. falciforme* predominated in DLSO, and variable appearance in PSO, TDO and endonyx onychomycosis [26]. The clinical differences between *Fusarium* and dermatophytes onychomycoses are not clear; however, some clues are available. Fusarium onychomycosis is most implicated in (1) periungual inflammation of the nail matrix and purulent discharge [4,9,27], (2) resistance to empirical antifungal treatment [25], (3) trauma history or nail dystrophy and absence of tinea pedis [28], and (4) involvement of the big toes (fingernails are only occasionally involved as combination symptoms) [23,24,29]. In the present study, there were six DLSO, two PSO (one overlapping paronychia), one WSO, and one DLSO phenotype with onychodystrophy from 10 cases. 

The histological presentation of Fusarium onychomycosis is only mentioned in the case of reports and is rarely systemically reviewed [30]. Lavorato et al. compared the performance of mycology and histology for dermatophyte and NDM onychomycoses and revealed that direct microscopy was more sensitive for NDM and that nail clippings for histopathology were better for dermatophyte onychomycosis [31]. However, this research only collected DLSO pattern onychomycosis, and only 28.5% cases were Fusarium onychomycosis. Although we cannot differentiate dermatophyte onychomycosis from NDM onychomycosis simply by histology, there may be additional clues. Among the six recognized patterns, the most frequently seen patterns in our study were frequently branching irregularly septated hyphae, moniliform hyphae, and hyphae with terminal swelling. Direct microscopic examination with the findings of chlamydospores with beaded appearance hyphae, terminal enlarging, and adventitious sporulation are also helpful for differentiation. 

The pathogenesis of *Fusarium’s* invasion of human nails has been previously elucidated [32]. In vitro, *Fusarium* species can destroy the stratum corneum through keratolysis without additional nutrients [32]. Further, marked protease activity has been detected in FSSC [33]. Flavia et al. demonstrated that *Fusarium oxysporum* invades nail plates, resulting in the ex-vitro formation of biofilm composed of hyphae, conidia, and extra matrix. The nail unit is a site of immune privilege with low expression of major histocompatibility antigens, dysfunction of antigen presenting cells, and inhibition of natural killer cell activity [34]. However, studies comparing NDM and dermatophyte onychomycoses in terms of levels of myotoxins, keratinase, and proteases along with components of fungal biofilms remain scarce. 

Prior to this study, most of our patients received combination (topical and systemic antifungals—TRB and ITC) or destructive (laser and surgery) therapy. Among them, 53.8% showed poor response to all treatment. In review articles, few treatment methods are listed, with 26.7% clinical and 13.9% mycological cure rates reported [8,25] Gupta et al. proposed a treatment algorithm for NDM onychomycosis using a combination of topical (EFC, Tavaborole, and LLC) and systemic (ITC and TRB) therapies [8], with ITC used as daily or pulse therapy (400 mg/pulse once a week for 3 weeks); both showed mild-to-moderate evidence of *Fusarium* clearance [35,36]. TRB is commonly used for onychomycosis, but the drug resistance rate is relatively high and requires combination with topical antifungals or keratolytics [37]. Verrier et al. reported that oral TRB and ITC are not effective in the treatment of Fusarium onychomycosis [38]. If treatment fails, an antifungal susceptibility test is indicated, and alternatives should be considered. There is one report of treating recalcitrant *Fusarium falciforme* with posaconazole pulse therapy (800 mg/pulse one week for each month, with total four months) in the literature; this reportedly achieved clinical and mycological improvement [39]. Combination therapy using topical EFC, oral ITC, and oral fosravuconazole are also approved for the treatment of onychomycosis in Japan [24]. Further, topical treatment with AMB for a year has shown a reduction in recalcitrant cases [40]. Other ablative treatment procedures such as two sessions of Qs Nd-YAG laser therapy (532 nm and 1064 nm) one month apart for patients with FSSC onychomycosis showed good response [41], while another study used 1340 nm laser monotherapy, resulting in persistent onychomycosis (91%) under mycological tests for one year [42]. Methylene blue-mediated photodynamic therapy is another choice, which may be superior to 5% amorolfine nail lacquer for NDM onychomycosis [43].

Currently, there are no clinical breakpoints for antifungal drugs against different *Fusarium* species. In this study, TRB, FLC, and ITC all showed poor improvement in treatment. However, AMB, EFC, LNC, LLC, VRC, and NAT showed better results and should be considered for clinical applications according to MIC results (Table 3). In the literature, The MIC levels from the north of Iran are compatible with our findings, as LLC and LNC was in the range of 1–0.001 μg/mL [26]. Based on Uemura et al., MIC in Fusarium onychomycosis, ITC, FLC, and 5-Fluorocytosine showed high resistance tendency; TRB showed variable resistance tendency [25]; and VRC and AMB showed low resistance tendency. The recently developed antifungal EFC has shown good treatment response in intractable cases [24,44], and olorofim has also shown promise as a candidate [45] after showing in vitro activity against FSSC and FOSC.

Identification of *Fusarium* to the species level is challenging for clinical laboratories as the morphological characteristics required for identification are few and require experience with the genus. Furthermore, the recent *Fusarium* taxonomy is based on molecular phylogeny, making identification by morphology alone unreliable. Among existing reviews and case studies on Fusarium onychomycosis, five identified the pathogen to the genus level and 11 to the *Fusarium* specie level; only five studies performed molecular identification to the species complex level [4,5,7,22,23,24,26,29,30,32,46,47,48,49,50,51,52,53,54]. Molecular identification is typically performed using phylogenetic analysis of sequences of ITS and *TEF-1α*, with RNA polymerase II’s second largest subunit (*RPB2*) genes sometimes used for better differentiation between species. Molecular identification can provide clues to the source and process of the infection, and the importance of repeated and accurate fungal culture reminds patients to pay attention to pathogen control in public health to identify and avoid the source of *Fusarium* colonization and invasion. Differences between clinical manifestation and antifungal susceptibility testing highlighted the importance of accurate molecular classification.

## 5. Conclusions

Although Fusarium onychomycosis accounts for 1–6% of cases of onychomycosis, its frequent resistance to treatment highlights its importance [5]. Cutaneous fusarium infections can serve as the origin of disseminated and invasive infection poor response to empirical antifungals. Therefore, accurate diagnosis through histological and molecular identification is required. Positive culture results for *Fusarium* species from nail samples are not a proof of infection; further histological proof or positive repeated culture results for the *same* species of *Fusarium* are required. Molecular identification (ITS + *TEF-1α*) and phylogenetic analysis can be applied for pathogen species confirmation. If a positive culture of *Fusarium* is simply due to colonization, then destruction of the colony and hygiene to prevent colonization is enough. However, if there is a true infection, then combination therapy (using topical and oral antifungal agents) and even surgical debridement are required. Dermatologists will do well to partner with mycologists specialized in *Fusarium* identification and application. In this article, we highlighted the clinicopathological features of Fusarium onychomycosis and provided six histopathological hints for differentiating Fusarium onychomycosis from dermatophyte onychomycosis. However, much work is needed to provide a standard effective treatment protocol for Fusarium onychomycosis.

This study has some limitations. The first is its design as a single-centre retrospective study. Additionally, owing to a lack of follow up with clinical photos, we could not determine and classify the percentage of accurate onychomycosis subtypes in this study. Last but not least, although current diagnostic criteria for Fusarium onychomycosis is not perfect, it is crucial for helping clinicians in diagnosis and treatment application.

## Figures and Tables

**Figure 1 jof-09-00534-f001:**
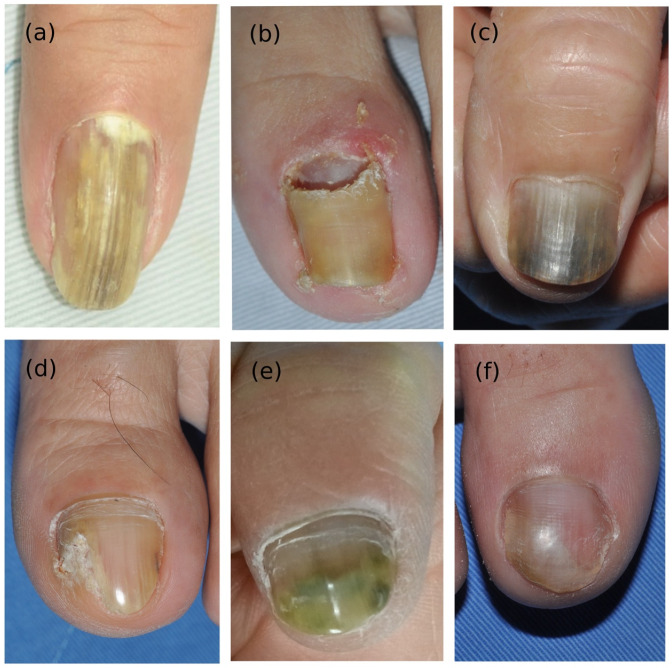
Clinical manifestations of Fusarium onychomycosis, including yellow and greenish discoloration (**a**,**b**,**e**), onycholysis (**a**–**f**), severe paronychia (**b**), and proximal subungual onychomycosis (**a**,**b**). The diversity of clinical presentation highlights the importance of fungus culture and species identification.

**Figure 2 jof-09-00534-f002:**
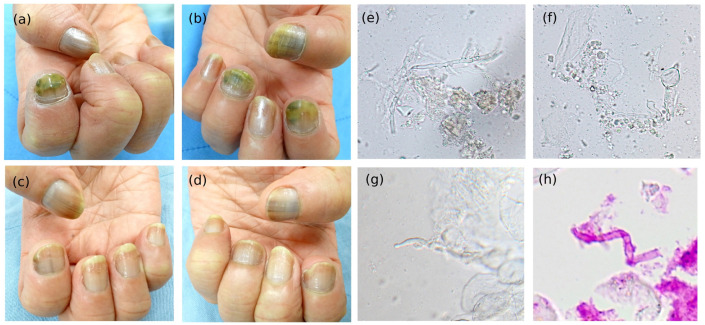
A 60-year-old female who had Fusarium onychomycosis on fingernails. *Fusarium keratoplasticum* was isolated from right 2nd fingernail and *F. solani* was isolated from left 1st, 2nd, and 4th fingernails. (**a**,**b**) Yellowish green discoloration on the right second finger, left thumb, and left second and fourth fingers for 2–3 months. (**c**,**d**) Three months after combination therapy with systemic terbinafine and topical sulconazole solution treatment; green discoloration of nails regressed with new nail regrowth. (**e**–**g**) Direct microscopic examination showing frequently branching irregularly septated hyphae, adventitious sporulation, and chlamydospore like swelling. (**h**) Sections showing twisted broad septated hyphae (Hematolysin and eosin stain, original magnification 200×).

**Figure 3 jof-09-00534-f003:**
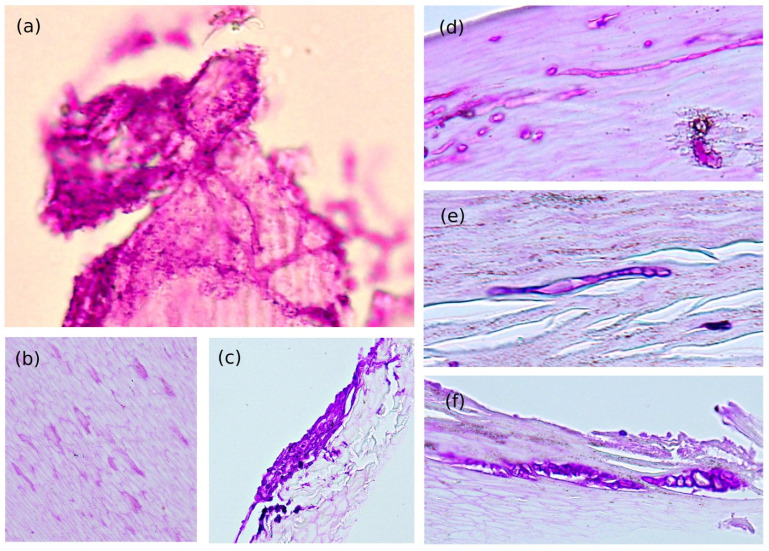
Histopathological characteristics of *Fusarium* onychomycosis classified into six subgroups. (**a**) Frequently branching irregularly septated hyphae, (**b**) Arbitrarily widening hyphae, (**c**) Dermatophytoma-like fungal mass, (**d**) Thin hyphae embedded in the nail specimen, (**e**) Moniliform hyphae, and (**f**) Hyphae with terminal swelling (Hematolysin and eosin stain, original magnification 200×).

**Figure 4 jof-09-00534-f004:**
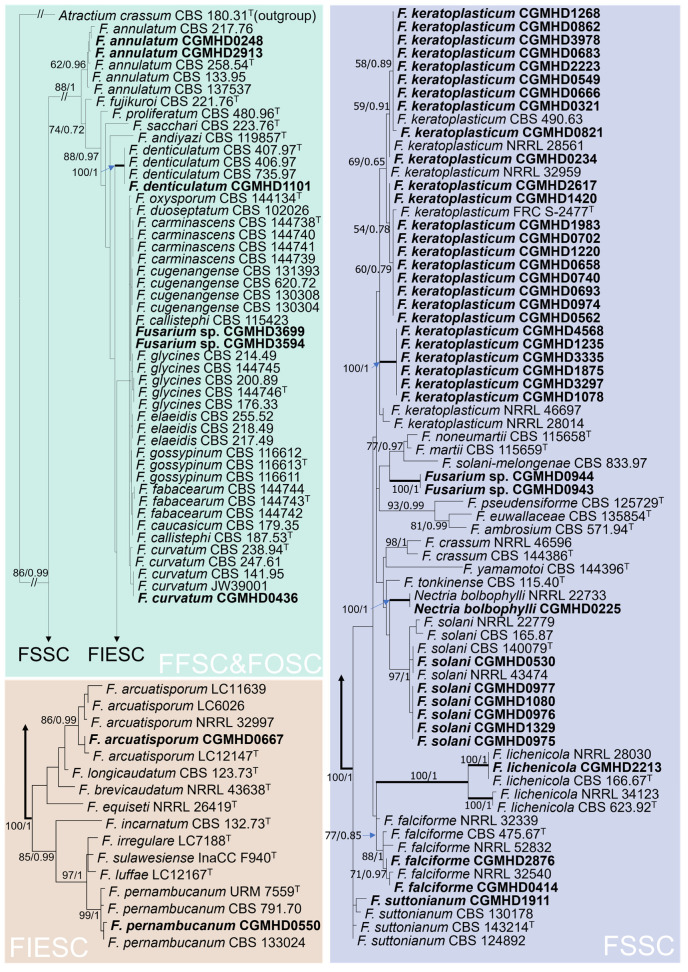
The maximum likelihood phylogenetic tree inferred from the concatenated TEF-1α and ITS regions. Bootstrap support value (ML) and Bayesian posterior probability (BP) higher than 50 and 0.6 are given at each node as ML/BP. Nodes with a support of 100/1.0 are shown in bold. The strains isolated in this study are shown in bold. Type strains are indicated with a superscripted T.

**Figure 5 jof-09-00534-f005:**
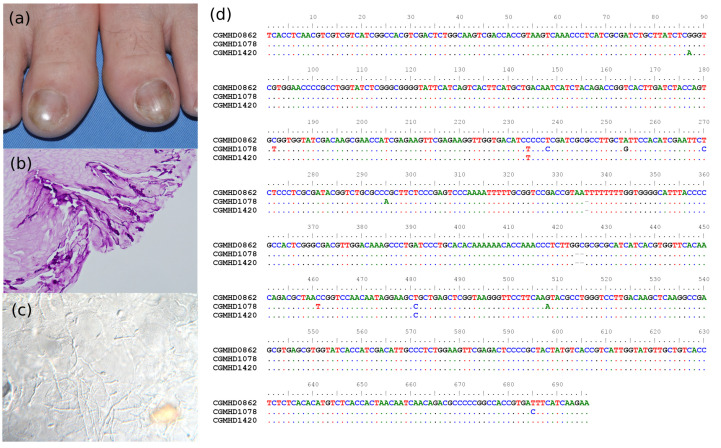
A 55-year-old female had Fusarium onychomycosis on her bilateral big toes. (**a**) Yellowish grey discoloration and onycholysis on the big toes. (**b**) Histopathology study showing septated and moniliform hyphae invasion into the nail specimen. (Hematoxysin and eosin stain, original magnification 200×) (**c**) Direct microscopic examination revealed septated hyphae and adventitious sporulation. (**d**) Transcription elongation factor-1α (TEF-1α) gene sequences of the isolates showed that the *F. keratoplasticum* in her toenails belonged to three different strain types: CGMHD0862 (left big toe), CGMHD1078 (right big toe), and CGMHD1420 (big toe, site not specified).

**Table 1 jof-09-00534-t001:** Clinical and mycological characteristics in 29 patients with Fusarium onychomycosis.

	Sex/Age	Immune Status	Location	Duration (Month)	Species	Treatment	Prognosis	Contact to Soil
**FFSC**	F/77	IC	Right big toenail	2	*Fusarium denticulatum*	TAF	GR	Yes
	F/61	IC	Right middle finger	1	*Fusarium annulatum*	Surgery	GR	Yes
	F/84	Lung cancer	Fingernails	6	*Fusarium annulatum*	ITC + TAF	GR	No
**FIESC**	F/65	Paraneoplastic pemphigus	Fingernails	108	*Fusarium pernambucanum* (FIESC 17)	ITC + TAF	PoR	No
	M/46	Tuberous sclerosis, left RCC	Toenails	3	*Fusarium arcuatisporum* (FIESC 7)	TAF	PaR	No
**FOSC**	F/54	IC	Right big toenail	53	*Fusarium curvatum*	TAF	Los	No
	M/36	IC	Bilateral big toenails	4	*Fusarium* sp.	ITC + TAF	GR	No
**FSSC**	F/60	SLE	Fingernails	7	*Fusarium keratoplasticum* (FSSC 2) *Fusarium solani* (FSSC 5)	ITC/TRB + TAF	GR	No
	M/21	IC	Left big toenail	4	*Fusarium solani* (FSSC 5)	TAF	PaR	No
	M/43	IC	Bilateral big toenail	1	*Fusarium falciforme* (FSSC 3 + 4)	TAF	PoR	No
	F/58	DM	Right big toenail	32	*Fusarium falciforme* (FSSC 3 + 4)	TRB + TAF	PoR	No
	F/45	IC	Left big toenail	4	*Fusarium keratoplasticum* (FSSC 2)	TRB +TAF	GR	No
	F/67	IC	Bilateral big toenail	60	*Fusarium keratoplasticum* (FSSC 2)	TRB + TAF	PoR	No
	M/77	IC	Bilateral thumb	48	*Fusarium keratoplasticum* (FSSC 2)	TRB +TAF	PoR	Yes
	F/3	IC	Fingernails + Toenails	15	*Fusarium keratoplasticum* (FSSC 2)	TAF	PoR	No
	F/31	IC	Left big toenail	32	*Fusarium keratoplasticum* (FSSC 2)	ITC/TRB + TAF	GR	No
	M/34	IC	Bilateral toenails	4	*Fusarium keratoplasticum* (FSSC 2)	TRB + TAF	GR	No
	F/79	IC	Fingernails + Toenails	25	*Fusarium keratoplasticum* (FSSC 2)	TAF	PoR	No
	F/55	IC	Bilateral big toenail	41	*Fusarium keratoplasticum* (FSSC 2)	Laser + Griseolfulbin + TAF	GR	No
	M/60	IC	Fingernails + Toenails	41	*Fusarium keratoplasticum* (FSSC 2)	TRB + TAF	PaR	No
	M/45	IC	Right big toenail	2	*Fusarium keratoplasticum* (FSSC 2)	TAF	PoR	No
	F/65	IC	Right big toenail	24	*Fusarium keratoplasticum* (FSSC 2)	TRB + TAF	PoR	No
	M/48	IC	Fingernails + Toenails	9	*Fusarium keratoplasticum* (FSSC 2)	TAF	PoR	No
	F/87	Terminal ileal cancer	Bilateral big toenail	36	*Fusarium keratoplasticum* (FSSC 2)	TAF	PoR	No
	M/70	DM	Left thumb	2	*Fusarium suttonianum* (FSSC 20)	TAF	GR	Yes
	M/77	IC	Fingernails + Toenails	6	*Fusarium lichenicola* (FSSC 16)	TAF	PoR	Yes
	M/65	IC	Right big toenail	5	*Nectria bolbophylli*	TRB + TAF	GR	No
	M/20	IC	Left 2nd fingernail	1	*Fusarium* sp.	TAF	Los	No
	F/61	IC	Fingernails + Toenails	5	*Fusarium* sp.	TRB + TAF	GR	Yes

Abbreviations: F: female; M: male; IC: immunocompetent; RCC: renal cell carcinoma; SLE: systemic lupus erythematosus; DM: diabetes mellitus; GR: Good response; PaR: Partial response; PoR: poor response; Los: Loss of follow up; TAF: Topical antifungals; ITC: Itraconazole; TRB: Terbinafine.

**Table 2 jof-09-00534-t002:** Six characteristic histopathology findings in Fusarium onychomycosis.

	(a) Frequently Branching Irregularly Septated Hyphae	(b) Arbitrarily Widening Hyphae	(c) Dermatophytoma Like Fungal Mass	(d) Thin Hyphae	(e) Moniliform Hyphae	(f) Hyphae with Terminal Swelling
**FSSC (N = 23)**						
*F. keratoplasticum*	4	1	3		7	4
*F. falciforme*	1				1	
*F. solani* SC	1					
*F. suttonianum*	1				1	
*F. lichenicola*	1					
*Nectria bolbophylli*	1					
*Fusarium* species	1		1		1	
**FFSC (N = 3)**						
*F. denticulatum*	1			1		
*F. annulatum*	2					
**FOSC (N = 2)**						
*F. curvatum*	1					
**FIESC (N = 2)**						
*F. pernambucanum*						1
*F. arcuatisporum*		1				
**Total**	14	2	4	1	10	5

**Table 3 jof-09-00534-t003:** The *Fusarium* species of clinical isolates and their minimum inhibitory concentration of 9 drugs (μg/mL) and GenBank accession numbers.

Species		RLMM No.										Accession Number	
			AMB	TRB	FLC	ITC	EFC	LNC	LLC	VRC	NAT	ITS	EF1a
FFSC	*F. annulatum*	CGMHD0248	1	4	>64	>32	0.25	0.063	0.031	4	4	LC687503	LC697741
	*F. annulatum*	CGMHD2913	2	4	>64	>32	0.5	0.125	0.063	4	8	LC687504	LC697742
	*F. denticulatum*	CGMHD1101	1	2	>64	4	0.125	0.031	<0.008	1	4	LC687507	LC697745
FIESC	*F. arcuatisporum*	CGMHD0667	ND	ND	ND	ND	ND	ND	ND	ND	ND	LC687505	LC697743
	*F. pernambucanum*	CGMHD0550	1	>32	>64	32	0.5	0.063	0.063	2	4	LC687537	LC697775
FOSC	*F. curvatum*	CGMHD0436	1	8	>64	>32	0.5	0.125	0.063	4	4	LC687506	LC697744
	*Fusarium* sp.	CGMHD3594	2	4	>64	>32	0.5	0.125	0.063	8	8	LC687546	LC697784
	*Fusarium* sp.	CGMHD3699	4	2	>64	>32	0.5	0.125	0.063	8	8	LC687547	LC697785
FSSC	*F. falciforme*	CGMHD0414	1	>32	>64	>32	1	0.5	0.25	8	8	LC687508	LC697746
	*F. falciforme*	CGMHD2876	2	>32	>64	>32	1	0.125	0.031	4	8	LC687509	LC697747
	*F. keratoplasticum*	CGMHD0234	2	>32	>64	>32	2	0.5	0.125	8	4	LC687510	LC697748
	*F. keratoplasticum*	CGMHD0321	1	>32	>64	>32	4	0.25	0.125	8	4	LC687511	LC697749
	*F. keratoplasticum*	CGMHD0549	2	>32	>64	>32	2	0.5	0.125	8	4	LC687512	LC697750
	*F. keratoplasticum*	CGMHD0562	>16	>32	>64	>32	>4	>4	>4	>16	>32	LC687513	LC697751
	*F. keratoplasticum*	CGMHD0658	2	>32	>64	>32	2	0.25	0.063	8	4	LC687514	LC697752
	*F. keratoplasticum*	CGMHD0666	2	>32	>64	>32	2	0.25	0.125	8	4	LC687515	LC697753
	*F. keratoplasticum*	CGMHD0683	2	>32	>64	>32	2	0.25	0.125	8	4	LC687516	LC697754
	*F. keratoplasticum*	CGMHD0693	2	>32	>64	>32	2	0.25	0.063	8	4	LC687517	LC697755
	*F. keratoplasticum*	CGMHD0702	1	>32	>64	>32	1	0.25	0.063	8	4	LC687518	LC697756
	*F. keratoplasticum*	CGMHD0740	4	>32	>64	>32	2	0.25	0.063	8	4	LC687519	LC697757
	*F. keratoplasticum*	CGMHD0821	4	>32	>64	>32	2	0.25	0.063	8	4	LC687520	LC697758
	*F. keratoplasticum*	CGMHD0862	2	>32	>64	>32	2	0.5	0.125	8	4	LC687521	LC697759
	*F. keratoplasticum*	CGMHD0974	>16	>32	>64	>32	>4	0.25	>4	>16	>32	LC687522	LC697760
	*F. keratoplasticum*	CGMHD1078	2	>32	>64	>32	2	0.5	0.125	8	4	LC687523	LC697761
	*F. keratoplasticum*	CGMHD1220	2	>32	>64	>32	2	0.25	0.063	8	4	LC687524	LC697762
	*F. keratoplasticum*	CGMHD1235	2	>32	>64	>32	0.5	0.125	0.031	2	4	LC687525	LC697763
	*F. keratoplasticum*	CGMHD1268	2	>32	>64	>32	2	1	0.125	16	4	LC687526	LC697764
	*F. keratoplasticum*	CGMHD1420	1	>32	>64	>32	2	0.25	0.063	8	4	LC687527	LC697765
	*F. keratoplasticum*	CGMHD1875	2	>32	>64	>32	0.5	0.125	0.031	4	4	LC687528	LC697766
	*F. keratoplasticum*	CGMHD1983	4	>32	>64	>32	2	0.25	0.063	16	4	LC687529	LC697767
	*F. keratoplasticum*	CGMHD2223	4	>32	>64	>32	2	0.25	0.063	8	4	LC687530	LC697768
	*F. keratoplasticum*	CGMHD2617	4	>32	>64	>32	2	0.25	0.063	8	4	LC687531	LC697769
	*F. keratoplasticum*	CGMHD3297	2	>32	>64	>32	0.5	0.125	0.063	2	4	LC687532	LC697770
	*F. keratoplasticum*	CGMHD3335	2	>32	>64	>32	1	0.25	0.063	4	4	LC687533	LC697771
	*F. keratoplasticum*	CGMHD3978	4	>32	>64	>32	2	0.25	0.063	8	4	LC687534	LC697772
	*F. keratoplasticum*	CGMHD4568	2	>32	>64	>32	1	0.25	0.063	8	4	LC687535	LC697773
	*F. lichenicola*	CGMHD2213	1	>32	>64	>32	0.5	0.125	0.063	4	16	LC687536	LC697774
	*F. solani*	CGMHD0530	0.5	>32	>64	>32	1	0.25	0.063	8	8	LC687538	LC697776
	*F. solani*	CGMHD0975	1	>32	>64	>32	2	0.5	0.063	8	4	LC687539	LC697777
	*F. solani*	CGMHD0976	1	>32	>64	>32	2	0.5	0.063	8	4	LC687540	LC697778
	*F. solani*	CGMHD0977	1	>32	>64	>32	2	0.5	0.063	8	4	LC687541	LC697779
	*F. solani*	CGMHD1080	1	>32	>64	>32	2	0.5	0.063	8	4	LC687542	LC697780
	*F. solani*	CGMHD1329	0.5	>32	>64	>32	1	0.5	0.125	8	4	LC687543	LC697781
	*Fusarium* sp.	CGMHD0943	>16	>32	>64	>32	>4	0.5	>4	>16	>32	LC687544	LC697782
	*Fusarium* sp.	CGMHD0944	4	32	>64	>32	0.5	0.125	0.031	4	8	LC687545	LC697783
	*F. suttonianum*	CGMHD1911	0.5	>32	>64	>32	1	0.5	0.25	4	8	LC687548	LC697786
	*Nectria bolbophylli*	CGMHD0225	4	>32	>64	>32	2	0.5	0.125	16	4	LC687549	LC697787

Abbreviations: RLMM: Research Laboratory of Medical Mycology, Chang Gung Memorial Hospital, Linkou Branch, Taoyuan, Taiwan; AMB: amphotericin B, TRB: terbinafine, FLC: fluconazole, ITC: itraconazole, EFC: efinaconazole, LNC: lanoconazole, LLC: luliconzole, VRC: voriconazole, NAT: natamycin; ND: not done due to loss of sporulation.

**Table 4 jof-09-00534-t004:** Treatment response and prognosis of Fusarium onychomycosis.

Treatment Methods	GR	PaR	PoR	Los
Itraconazole + topical antifungals (N = 2)	1	0	1	0
Terbinafine + topical antifungals (N = 9)	3	1	5	0
Itraconazole/Terbinafine + topical antifungals (N = 3)	3	0	0	0
Topical antifungals only (N = 13)	2	2	7	2
Laser + Griseofulvin + topical antifungals (N = 1)	1	0	0	0
Surgery (N = 1)	1	0	0	0
Total number (N = 29)	11	3	13	2

Abbreviations: GR: Good response; PaR: Partial response; PoR: poor response; Los: Loss of follow up.

## Data Availability

The data presented in the manuscript are available on request from the corresponding authors.

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
