# Peer review of "High Diversity of Fusarium Species in Onychomycosis: Clinical Presentations, Molecular Identification, and Antifungal Susceptibility"

_jof, 2023, doi:10.3390/jof9050534_

Round 1

Reviewer 1 Report

Globally, this work is good. The context, especially the clinical questions related to these molds misidentification, is clearly explained.

The references used are relevant and techniques used are adequate considering the objectives of the study.

Authors show the importance of identifying to which species or complexes of species a strain belongs, especially with regard to the levels of sensitivity to antifungal agents.

Several modifications are necessary:

Lines 56-57: The introduction would lack references for factors affecting epidemiological profiles.

 Line 72-79: Reference 8 is not valid for this section, I think the correct reference is 7. Please check that there are no other reference inversions in the document.

 Line 99: The reference article number 9 refers to the IQ-TREE phylogenetic package and not the DDJB.

 Line 103: I can't access the Fusarium Database (http://isolate.fusariumdb.org). Perhaps it’s due to a too restrictive setting of my university.

 Section 3.2: “chlamydospore-like swelling (Figure S1f)”   This is on figure 2f.

 Figure 2: In the legend, there is a typographical mistake, you write “chlamodospore”

 Section 3.4: “FSSC had a higher TRB MICs than non-FSSC.”, FSSC and FIESC although there is CLSI data for only one strain in this group.

 Section 3.6, case 2: “Molecular identification proved that all of them are F. keratoplasticum, but of three different genotypes based on the TEF-1α sequences (Figure 5d)”. This assertion is not supported by the quoted figure. Figure 5d show two genotypes only: One for CGMHD0862 and CGMHD1268 strains and one for CGMHD1078 strain like described in legend.

 In discussion, “PSO, TDO, and SWO phenotypes according to Uemura EVG et al, 24”: reference’s number is incorrectly dyctalographed.

Same remarks in this paragraph : “The MIC levels from North of Iran are compatible with our findings, as LLC and LNC was in the range of 1-0.001 μg/ml [25]. Based on Uemura EVG et al, MIC in Fusarium onychomycosis, ITC, FLC, and 5-Fluorocytosine showed high resistance tendency, TRB showed variable resistance tendency, and VRC and AMB showed low resistance tendency.24 The recently developed antifungal EFC has shown good treatment response in intractable cases [23 43], and olorofim has also shown promise as a candidate [44] after showing in vitro activity against FSSC and FOSC.”

You must verify all references because several problems were detected (references inversion, etc.).

Reviewer 2 Report

This is an interesting work, taking into account the many inconveniences that exist regading the NDM onychomycosis diagnosis.

Some parragraphs must be edited for better understanding.

Reviewer 3 Report

This is an interesting manuscript on Fusarium species in onychomycosis.  I advice to not report a table with all cases on onychomycosis (Table 1) but a unique summary table with location, duration, species, treatments, prognosis to make clear the presentation of data. 
